# GIS-Based Spatial Analysis of Accident Hotspots: A Nigerian Case Study

**Abayomi Afolayan** [1], **Said M. Easa** [2,*], **Oladapo S. Abiola** [1], **Funmilayo M. Alayaki** [1] **and Olusegun Folorunso** [3]

1 Department of Civil Engineering, Federal University of Agriculture, Abeokuta PMB 2240, Nigeria
2 Depatrment of Civil Engineering, Toronto Metropolitan University, Toronto, ON M5B 2K3, Canada
3 Department of Computer Science, Federal University of Agriculture, Abeokuta PMB 2240, Nigeria
* Correspondence: seasa@ryerson.ca; Tel.: +1-41-69-795-000 (ext. 7868)

**Abstract:** This study identified high-risk locations (hotspots) using geographic information systems (GIS) and spatial analysis. Five years of accident data (2013–2017) for the Lokoja-Abuja-Kaduna highway in Nigeria were used. The accident concentration analysis was conducted using the mean center analysis and Kernel density estimation method. These locations were further verified using Moran's I statistics (spatial autocorrelation) to determine their clustering with statistical significance. Fishnet polygon and network spatial weight matrix approaches of the Getis–Ord Gi* statistic were used in the hotspot analysis. Hotspots exist for 2013, 2014, and 2017 with a significance level between 95–99%. However, hotspots for 2015 and 2016 have a low significance level and the pattern is random. The spatial autocorrelation analysis of the overall accident locations and the Moran's I statistic showed that the distribution of the accidents on the study route is random. Thus, preventive measures for hotspot locations should be based on a yearly hotspot analysis. The average daily traffic values of 31,270 and 16,303 were obtained for the northbound and southbound directions of the Abaji–Abuja section. The results show that hotspot locations with high confidence levels are at points where there are geometric features.

**Keywords:** accidents; geographic information system; highway; hotspots; identification



## 1. Introduction

Globally, the transportation challenges faced by various nations have significantly increased. This increase has necessitated a search for methods that ensure efficient, safe, feasible, and faster means of transportation [1]. Transportation is vital to economic success and the quality of life in urban and rural areas. However, the growth of city populations, transportation infrastructure, and the corresponding distance travelled have generated adverse effects, such as congestion, air pollution, noise pollution, and motor vehicle collisions. An accident is unpalatable damage that occurs suddenly without knowing. Road accidents are a menace to the safety of families and they are associated with many problems that need to be treated individually as road, human, vehicle, and environmental factors play roles before, during, and after an incident [2]. Road traffic accidents happen when a vehicle collides with another vehicle, pedestrian, animal, road debris, or stationary objects, such as a tree or a utility pole [3].

A hotspot refers to a location along the road that is considered high-risk for vehicle collisions. Elvik [4] presented a conceptual meaning of a hotspot road section as any section with more expected accidents than other corresponding sections due to peculiar hazard factors prevalent in the section. He further outlined seven criteria of a modern hotspot identification method as: (a) identification of hazardous road locations from the population of sites, (b) avoidance of the sliding window method in hazardous road location identification, (c) use of the empirical Bayes (EB) method of hazardous road location

identification based on the expected number of accidents at a particular site, (d) in a population of sites, hazardous road locations should be identified as the upper limit of the EB distribution estimation, (e) a short period (3–5 years) of data is appropriate for the identification of hazardous locations and the development of an accident prediction model, (f) on the condition that the EB estimates of the expected number of accidents by severity for a particular site can be determined, accident severity can be taken into account when determining hazardous road locations, and (g) particular types of accidents can be looked into when determining hazardous road locations, on the condition that EB estimates of the expected number of accidents of the specific type can be acquired for the particular site. Hotspot programs are planned to reduce the collision risk in areas by improving the physical conditions or management [5]. According to [6], the hotspot is the number of personal injury accidents occurring within a 100 m grid square or 100 m length in three years in a particular road class. Therefore, the area is deemed a high-risk site if 20 accidents are recorded over three years on a 100 m length of road.

Overgaard Madsen [7] gave four criteria that a definition of hotspot location must satisfy: (a) random fluctuations in the number of accidents should be controlled, (b) factors responsible for having an impact on road safety should be considered, (c) sites with an overestimation of fatal and severe injury accidents should be identified, and (d) locations at which the local hazard factor associated with road design and traffic control made a considerable contribution to accident occurrences should be determined. These highway network spots are targeted at an all-inclusive safety program by traffic officials. The most prevalent challenges traffic officials face surround where and how to enforce preventive measures and provisions to maximize traffic safety [8].

The geographic information system (GIS) is a comprehensive management tool for traffic safety. The system has several benefits: (a) it allows managers to retain a large amount of data that can be easily stored, shared, and managed, (b) it enables a platform for data analysis and visualization to examine affinity between data, and (c) it can provide graphical and non-graphical results. Due to the spatially distributed nature of accidents, the use of GIS provides the capability to store, update, retrieve, compare, and spatially display data [9]. GIS allows hotspot maps to be electronically generated from a well-designed accident database and produce high accident rankings based on the total accidents occurring or accident rates. The advances in GIS and remote sensing can be effective for accident analysis. In addition to promoting the linkage between various types of data and maps, GIS can visually display the results of analyses, thus enabling sophisticated analysis and quick decision-making. Also, these tools would make the analysis less time-consuming and less tedious. Thus, GIS offers a platform to maintain and update the accident record database, which can be used for further analysis [10]. Hence, there is a need to apply these tools for the analysis of the hotspots along the Lokoja−Abuja−Kaduna highway.

All efforts to reduce the effects of traffic collisions are critical. Amidst these, identifying hotspots and considering likely causes have been studied extensively. Hotspot identification is usually the first step in a safety improvement program. In many safety improvement programs, sites are ranked according to their conditions and a subset of sites are then selected as the highest accident risk sites. Since budgets are limited, priority is given to these high-risk locations to implement preventive measures [11].

Road rehabilitation funds are often misappropriated as a site requiring adequate maintenance is sometimes neglected. Hence, there is a need to identify high-risk locations that require urgent maintenance works, thus allowing for the proper application of road maintenance funds. In this study, spatial analysis was selected based on its ability to detect sections having a higher number of accidents compared to other similar locations. Furthermore, it allows for the spatial dependence of collisions and helps identify segments with a significant spatial correlation that requires further analysis and safety [12]. This is more beneficial compared to the generalized linear models, which can only provide the relation between covariates and response in a linear additive manner.

The Lokoja−Abuja−Kaduna highway is a major federal highway that connects the northwest and north central zone to the southwest zone of Nigeria. Especially in the festive periods, the number of vehicles driving the route is dramatically increased compared to other periods. Hence, traffic accidents have become more common. Thus, it becomes necessary to reduce these accidents by conducting a comprehensive analysis and taking precautions. This study uses GIS to identify the hotspots along the Lokoja–Abuja–Kaduna highway in Nigeria. The study involved producing hotspot maps, classifying them based on density and confidence level, and examining their roadway geometric features to determine how, what, and where accident countermeasures can be applied.

The main contribution of this study is to demonstrate the viability of the fishnet polygon and spatial weight matrix in identifying accident locations and conceptualizing the spatial relationship among the locations on a highway network. The aim is to identify high-risk locations that require urgent maintenance work. For example, this approach uses the distance between features within the network, not the ordinary Euclidean distances used in the literature. Also, the generated network spatial weight matrix is fed into the Getis–Ord Gi* (GOG) statistic instead of the bandwidth implemented in previous studies.

The remaining sections of the paper are organized as follows: Section 2 presents a review of the spatial analysis methods of accident hotspots; Section 3 describes the data collection and GIS analysis; Sections 4 and 5 present the results and discussion, respectively; The conclusions are presented in Section 6.

## 2. Review of Spatial Analysis Methods

Spatial data and analysis are some of the most essential information for traffic accident analysis. GIS-aided spatial data and spatial analysis provide factual information to analysts about dangerous locations, hotspots, and warm spots. With GIS, the analyst can combine accident and highway data, geocode the accident data and locations, calculate the frequency and rate of accidents, and select a variable for stratification to calculate the mean and standard deviation of accident rates [13]. Identification of defective safety locations with GIS-aided spatial analysis will help to reduce traffic accidents. However, the success of these analyses relies solely on the precision, reliability, and all-inclusiveness of the traffic accident data. Countries are not in agreement on items that should be included in the traffic accident reports [14]. Aderinlewo and Afolayan [15] developed road accident prediction models for the Akure−Owo highway, Ondo State, Nigeria, based on field surveys and the Nigerian Federal Road Safety Commission (FRSC) accident reports. They found that the FRSC report forms were not detailed enough about accident occurrences at the locations along the study route. In addition, there were discrepancies amid the accident data of different years regarding the parameters included in the report.

Since 1990, various researchers have studied GIS technologies and their applications in the spatial pattern of accident analysis. These cut across spatial accident analysis models, spatial query, pattern analysis, proximity analysis, and segment and intersection analysis. The effects of various factors on safety performance are examined through traffic safety studies. These include the influence of geometric features of road design, environmental factors (e.g., weather conditions), and geographic conditions on accident occurrences [8,16–18].

Easa and Chan [19] presented various GIS applications for urban planning and development, including transportation, public utilities, remote sensing, trends in spatial databases, linear referencing systems, demographic forecasting, stormwater and waste management, and environmental assessment of air quality. Aguero-Valverde and Jovanis [12] investigated the effect of spatial correlation in models of road accident frequency at the segment level. The study revealed that spatial correlation models better fit the data than the Poisson-lognormal model consisting of different or diverse elements. Owusu et al. [20] analyzed a road traffic accident pattern in the Cape Coast Metropolis of Southern Ghana using GIS. Sandhu et al. [21] identified highway hotspots using the Kernel density estimation (KDE) method, where GIS was used to map, visualize, and examine

accident data. The hotspots were verified using the Getis–Ord Gi* and Global Moran's I statistics to measure the spatial autocorrelation.

Moran [22] discussed the tests of significance for the random distribution of some quality or phenomenon in a country or state to ascertain whether the factors causing the events can be taken as statistically independent in different countries or not. A standardized distribution was shown to tend to normality for the events. In a pilot study, Ref. [11] determined traffic accident hotspots on the Turkish highway road network by comparing the traditional hotspot detection methods with the spatial statistical methods. The spatial methods were susceptible to accidents that occurred involving multiple vehicles. In a further study, Ref. [23] used GIS as a management system for accident analysis and statistical analysis to determine accident hotspots in the Afyonkarahisar administrative border in Turkey. They inferred that traffic agencies could retrieve, analyze, and display accident data in a correctly set up GIS system. Olusina and Ajanaku [24] also mapped accident hotspots from primary and secondary data sources. The accident spot severity and venerability were determined based on the weighted severity index using KDE methods. Verma and Khan [25] also identified the most vulnerable accident hotspots along Sagar–Shahgarh districts using a weighted severity index. The cluster analysis was conducted using spatial autocorrelation to ascertain the level of distribution of the hotspots. It was concluded that the research could be a vital tool for stakeholders in the road transportation sector. Getis [26] shows that spatial interaction models are a unique case of a common model of spatial autocorrelation. In the study, several conventional standards of spatial autocorrelation were indicated to possess a cross-product form. This was achieved by developing a spatial autocorrelation statistic that also doubles as a measure of spatial interaction. Sabel et al. [27] used KDE cluster analysis to identify road accident hotspots in Christchurch, New Zealand. Bello [28] also examined a stratified accident analysis in the city of Richardson using kernel densities. In Honolulu, Hawaii, spatial patterns of pedestrian accidents were analyzed by Kim and Yamashita [29] and Levine et al. [30] using the k-means clustering method. Sajed et al. [31] combined accident data, traffic, and geometric characteristics to identify hotspots.

## 2.1. Comparison of Various Methods of Hotspot Analysis

Considerable advances have been made in hotspot identification on the roads throughout the last few years. This was made possible by GIS and global positioning system (GPS) applications in transportation research. Various hotspot identification methods have been used in the literature, including global indices such as GOG, Geary's C, and Global Moran's I (spatial autocorrelation). Also used in the literature are local indices, such as Kriging, Local Anselin Moran's I, KDE, spatial analysis along network (SANET), KDE+, and spatial traffic accident analysis (STAA). Except for Kriging and KDE, these methods test the statistical significance of accident clusters [32]. The number of events over a unit area at a specified location (i.e., first-order properties) is addressed using KDE in a spatial hotspot analysis of point (point pattern analysis). In contrast, the second-order properties are addressed by Geary's C, GOG, and Moran's I, which deal with the spatial dependence and statistically evaluate the interaction between several events in pairs in a specified area [33]. Kriging is an improved spatial analysis approach primarily used in various research fields [34,35]. The SANET toolbox is used to overcome the shortcomings of planar spatial analysis for point events that are restricted to linear networks [36]. This toolbox is a spatial network analysis that evaluates the intensities of points on a network and outlines the network sections with high intensities. Compared to the planar spatial analysis method, it is highly efficient for a network with Euclidean distances which are prone to error [32,36,37]. The STAA method is a hazard-based approach that considers accident severity, frequency, and socioeconomic influences to analyze historical accident data [38]. STAA is a network-defined method comparable to SANET−KDE and KDE+. However, unlike SANET–KDE, STAA demands that the accident points overlap with the road centerline. By so doing, the initial coordinates of the accident points are maintained. This method is appropriate for analyzing single roads and networks of roads [32].

The methods used in the cluster analyses of road traffic accidents are the K-function, nearest-neighbor, KDE, dangerousness index (DI), hierarchical clustering (HC), and climbing. The K-function and nearest-neighbor methods provide evidence about the tendency of clustering on a road section but cannot specify the specific part of the section it occurred. Therefore, these methods do not contribute to the clusters' localization within the section [39]. The KDE and DI methods can identify the actual cluster position within a section or a network. The HC method has no mechanism for determining the statistical significance of clusters. It could only recognize the clusters of traffic accidents. The DI method is a particular case of KDE and relies on the 'points of measurement.' The climbing method can determine the cluster positions but it is highly susceptible, implying that a small change in the location of road traffic accidents outside of a cluster can substantially influence the cluster's importance.

As stated by Erdogan et al. [23], Sabel et al. [27], Anderson [40], and Plug et al. [41], the KDE, cluster analysis, and GOG are among the most effective and frequently used methods for the identification of the actual cluster location (hotspot) within a network or section. The KDE approach's principal merit is that the kernel's bandwidth is used to express the uncertainty about the actual accident location. This implies that KDE allows for the spreading of the risk of an accident [40]. As [41] indicated, KDE is more suitable for visualization than identifying hotspots. Presently, an inclusive examination of the statistical significance of KDE is lacking in the literature. Network KDE is more efficient for analyzing accidents on a 1D linear space (e.g., a road) [42–44]. An extended KDE approach evaluates the probability density function of the event points using the kernel function KDE+. The + signifies the criticality in the selection of significant clusters [45]. However, the approach is limited in that it is very effective for event points along the segments between the intersections since many accidents at intersections can surpass the occurrence of hazardous locations at the road segments between the intersections [39,46].

Several studies have used planar spatial analysis methods, such as KDE, Kriging, Local Anselin Moran's I, and GOG for hotspot analysis with global indices, including Global Moran's I and Geary's C. However, only a few studies compared the different approaches to hotspot identification. Thus, this paper used the two approaches (fishnet polygon and network spatial weight matrix) to identify hotspot locations in the study area.

In these approaches, the distances between the network features were measured as adopted in the literature and these were not the ordinary Euclidean distances. Also, the generated network spatial weight matrix was fed into the GOG statistic instead of the bandwidth.

### 2.2. Theoretical Analysis

Various studies on GIS applications, such as Erdogan et al. [23], Deepthi and Ganeshkumar, 2010 [10] have comprehensively covered the GIS and spatial analysis of accident hotspots. Thus, this paper only describes equations that explain the key parameters and their significance.

### 2.2.1. Mean Center Analysis

This method involved measuring the possible geographic mean of the accident locations along the highway network, taking the frequency of accidents at sites as a weight. The weighted mean center algorithm pulls the geographic center value or frequency toward accident locations with higher frequency attributes. The output of this computation can give the analyst an idea of where more accidents are concentrated in the study area. The mean center and weighted mean center are given by

$$\dot{X} = \frac{\sum_{i=1}^n x_i}{n}, \; \bar{Y} = \frac{\sum_{i=1}^n y_i}{n} \tag{1}$$

$$\dot{X}_w = \frac{\sum_{i=1}^n w_i x_i}{\sum_{i=1}^n w_i}, \; \bar{Y}_w = \frac{\sum_{i=1}^n w_i y_i}{\sum_{i=1}^n w_i} \tag{2}$$

where $x_i$ and $y_i$ are the coordinates of feature $i$, $n$ is the total number of features, and $w_i$ is the weight of feature $i$.

### 2.2.2. Kernel Density Estimation

The kernel density estimation was performed on the data to generate a subjective heat surface of the variation in the values of traffic accidents from high to low. This measure estimates the proportion of the total accidents that can be expected to occur at any given map location. It provides an estimate of the proportion of the total accidents that can be expected to occur in any given map location. The kernel density estimation is given by Equation (3) [10].

$$f(x, y) = \frac{1}{nh^2} \sum_{i=1} K\left(\frac{d_i}{h}\right) \tag{3}$$

where $f$ is the density estimate at location $(x, y)$, h is the search radius (bandwidth or kernel size), $n$ is number of observations (total number of accidents), $K$ is the kernel function, and $d_i$ is the distance between the location (event point) $(x, y)$ and the location of the $i$th observation.

The search radius, $R$, is given by

$$R = 0.9 \min\left(SD, \sqrt{\frac{1}{In(2)}} D_m\right) * n^{-0.2} \tag{4}$$

where $SD$ is the standard distance, $D_m$ is the median distance, and $n$ is the number of points (if no population field is used) or the sum of the population field values (if a population field is supplied).

### 2.2.3. Cluster Analysis

The spatial autocorrelation (Moran's $I$) algorithm simultaneously measures both features' locations and values and returns the pattern expressed by the data regarding whether they are clustered, dispersed, or random. Moran's I is an inferential statistical method, which means the analysis results are interpreted within the null hypothesis. This analysis was done for the individual years to observe whether there are changes in cluster intensity. Accident locations with a very low or very high $Z$-score fall outside the normal distribution and indicate a statistically significant area for analysis. The Moran's $I$ statistic, $I$, is given by

$$I = \frac{n \sum_{i=1}^{n} \sum_{j=1}^{n} w_{i,j} z_i z_j}{S_o \sum_{i=1}^{n} z_i^2} \tag{5}$$

where $z_i$ is the deviation of an attribute for feature $i$ from its mean $(xi - X)$, $w_{(i, j)}$ is the spatial weight between feature $I$ and $j$, $n$ is equal to the total number of features. Therefore, the aggregate of all the spatial weights is given by

$$S_o = \sum_{i=1}^{n} \sum_{j=1}^{n} w_{i,j} \tag{6}$$

The $Z$-score for the statistic is given by

$$Z = \frac{I - E[I]}{\sqrt{V[I]}} \tag{7}$$

where

$$E[I] = -\frac{1}{n-1} \tag{8}$$

$$V[I] = E[I^2] - E[I]^2 \tag{9}$$

2.2.4. Hotspot Analysis

This analysis was conducted in two ways: fishnet polygon analysis and network spatial weight matrix analysis. The fishnet polygon analysis aggregated accident locations into a fishnet grid. Each grid contained several accident locations that represent the weight of the grid. The GOG statistic is given by

$$G_i^* = \frac{\sum_{j=1}^n w_{i,j} x_j - \dot{X} \sum_{j=1}^n w_{i,j}}{S \sqrt{\frac{[n \sum_{j=1}^n w_{i,j}^2 - \left(\sum_{j=1}^n w_{i,j}\right)^2]}{n-1}}} \tag{10}$$

where

$$\dot{X} = \frac{\sum_{j=1}^n x_j}{n} \tag{11}$$

$$S = \sqrt{\frac{\sum_{j=1}^n x_j^2}{n} - \left(\dot{X}\right)^2} \tag{12}$$

where $x_j$ is the attribute value for feature $j$, $w_{i,j}$ is the spatial weight between feature $i$ and $j$, and $n$ is equal to the total number of features.

## 3. Data Collection and Analysis

### 3.1. Study Route

The study route is the Lokoja−Abuja−Kaduna highway. It lies between latitudes 07°47′ N, 09°05′ N, and 10°30′ N and longitudes 06°45′ E, 07°32′ E, and 07°21′ E, respectively. There are three state capitals connected to the route. Lokoja is located in the north central zone of Nigeria, the capital of Kogi state. Abuja is located in the north central of Nigeria and it is the capital of Nigeria. In contrast, Kaduna is located in the northwest of Nigeria and it is the capital of Kaduna State. The Lokoja to Kaduna highway is a dual carriageway. The section in Lokoja starts at the Lokoja central market intersection and ends at the Abuja intersection for the section in Kaduna. The total length of the study section is 385 km. The accident data for 2013–2017 were obtained from the FRSC.

A preliminary analysis determined the number of accident occurrences at locations along the route in the year under consideration. This step was conducted through the GIS map, where the names of places were indicated along the route. Figure 1 shows the GIS map for the study area.

The map shows spatial autocorrelation as most neighboring locations along the route display similar configurations. This spatial relationship is further analyzed through appropriate tests in the next section.

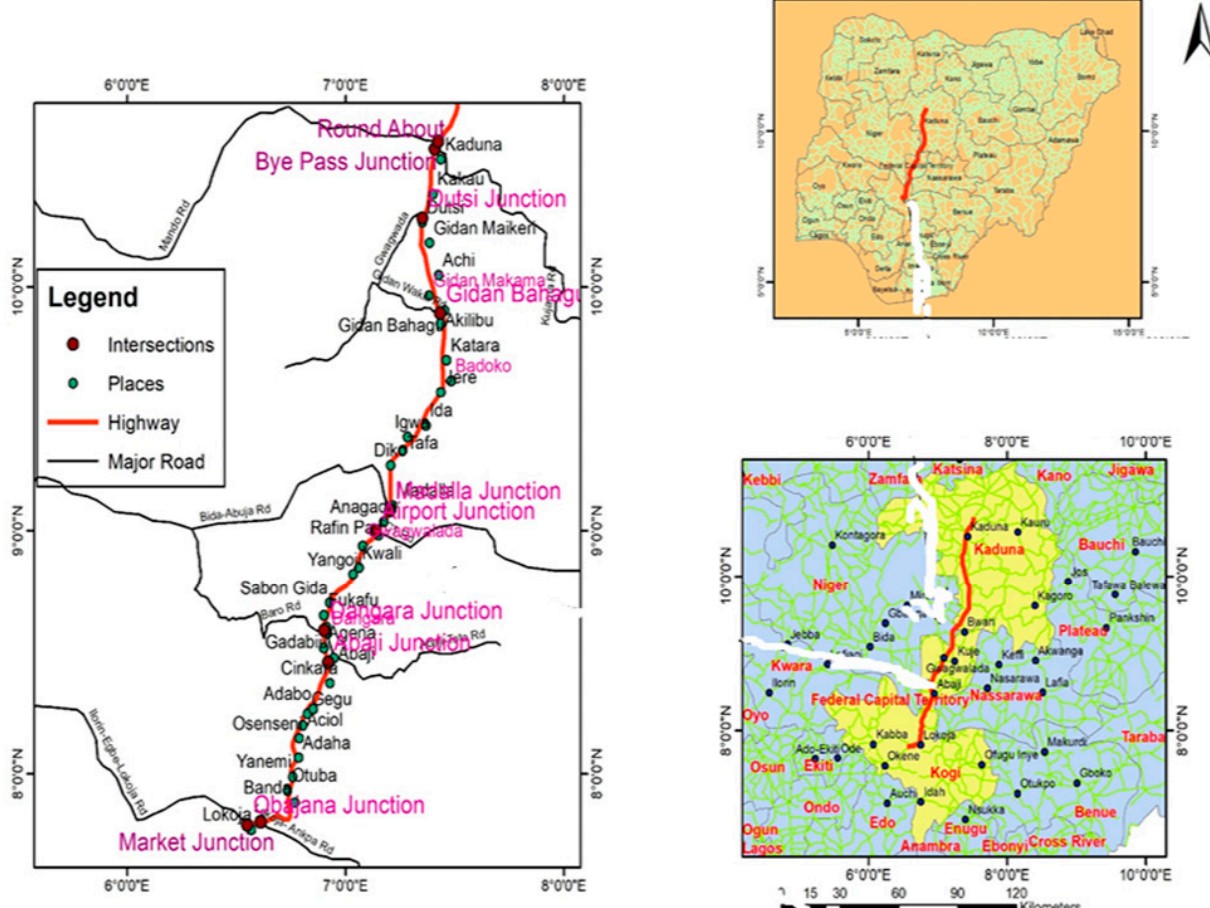

**Figure 1.** GIS map of the study route.

### 3.2. Data Collection

A desk study was conducted for the accident data acquired from the FRSC Abuja office. The locations along the study area where accidents had occurred from 2013 to 2017 were extracted and tabulated for further analysis. A reconnaissance survey was conducted on the study highway. The FRSC unit command along Lokoja-Abuja-Kaduna highway Nigeria and accident emergency response (ZEBRA) along the route were contacted for information regarding accident occurrence. Data were acquired from both primary and secondary sources. The primary data source included geometric (coordinates) and attribute data of the accident spots. The field data were collected using Garmin-handled GPS, where the GPS survey was conducted for all the locations of the accident points. The secondary data source included Google Earth imagery, accident data from the FRSC headquarters Abuja, Nigeria, and records of online accident reports. These data aided in identifying accident locations quickly and gathering information promptly. The Google Earth interface covering the study area was imported into the ArcGIS 10.2.1 environment using the Arc2Earth extension tool. Features such as the study route, intersection, U-turns, and intersecting roads were digitized using the Editor.

A traffic counter was installed at the study area to obtain the traffic volumes for the sections with hotspot locations. In addition, a 24-h count for seven days was conducted and the average daily traffic (ADT) was obtained for the sections. These data were needed to determine the relationship between the number of accidents and the traffic volume. In addition, the data helped to determine the significance of traffic exposure on accident occurrence at the hotspot locations.

*3.3. GIS-Based Analysis*

The causes of accidents from the record of the FRSC were summarized on maps, charts, and tables. To validate the accident records from the FRSC and identify hotspots along the study route, four different types of analysis executed in ArcGIS were conducted: mean center analysis, KDE, cluster analysis, and hotspot analysis.

*Mean Center Analysis*: This method measured the possible geographic mean of the accident locations along the highway network, taking the accident's frequency as a weight. The weighted mean center algorithm pulls the geographic center value or frequency toward accident locations with higher frequency attributes. The output of this computation can give the analyst an idea of where more accidents are concentrated in the study area. Specifically, the calculation was done for each year and displayed in a single window to show any noticeable shift in the mean center.

*Kernel Density Estimation*: The kernel density estimation was performed on the data to generate a subjective heat surface of the variation in the values of traffic accidents from high to low. This measure estimates the proportion of the total accidents that can be expected to occur at any given map location and intersection.

*Cluster Analysis*: the spatial autocorrelation (Moran's *I*) algorithm simultaneously measures both features' locations and values. It returns the pattern expressed by the data regarding whether they are clustered, dispersed, or random. Moran's I is an inferential statistical method, which means the analysis results are interpreted within the null hypothesis. The null hypothesis assumes complete spatial randomization. That is, values are randomly distributed among features, reflecting a random spatial process at work. This method was intended to reveal whether the clusters with high or low traffic accident values are statistically significant. Moran's *I* was referred to as the high or low cluster (Getis Ord. General G) because the values associated with the accident points are not reasonably evenly distributed (as examined using KDE) across the study area. General G statistics are more appropriate for achieving such a distribution, where the local spikes in the values are picked as clusters of high values. Moran's *I*, however, is suitable to look at clusters in the dataset because it correlates the feature values globally with a fixed distance band or the average nearest neighbor and returns the features as clustered, dispersed, or randomly distributed.

For this analysis, a network spatial weight matrix that chooses the eight nearest neighbors and a distance on a network of 7 km was generated. The scale of the analysis indicates this is a road stretching 385 km and shows the separation between accident locations. A smaller neighborhood selection would result in many of the features not having neighbors in the analysis, resulting in outputs that are not representative of the phenomena in the study route. This is consistent with the recommendations of [36] regarding bandwidth to cell width. Chainey [47] also inferred that the selection of widths should be as effective as possible to guarantee that visual appeals and spatial patterns do not jeopardize the precision of output results. This analysis was done for the individual years to observe whether there are changes in cluster intensity. The **Z**-scores for each year were plotted against the corresponding year to create a line chart. The **Z**-score is a standard deviation measure for each of the 88 locations. Accident locations with a very low or very high **Z**-score fall outside the normal distribution and indicate a statistically significant area for analysis.

*Hotspot analysis*: This analysis was conducted using fishnet polygon analysis and network spatial weight matrix analysis. The fishnet polygon analysis aggregated accident locations into a fishnet grid. Each grid contained several accident locations that represent the weight of the grid. An optimized cell size of 4.239 km was used as the fishnet polygon mesh size for aggregating incidents. This was considered adequate because each grid contained at least one accident point for the chosen cell size. There were 48 weighted polygons with a weighted mean of 1.8333, a minimum of 1.00, and a maximum of 7.00. An optimized average distance to the 3 nearest neighbors for each fishnet polygon was used for the analysis.

The network spatial weight matrix was used to conceptualize the spatial relationship among the accident locations on the highway network. This was the input instead of

the bandwidth into the GOG statistic for the algorithm to respect the peculiar network distance between the highway features. This is similar to conducting the SANET tool used by Zahran et al. [32]. The matrix was generated on a network dataset comprising highway features and intersections for this analysis. Therefore, the distances between the features were measured within the network and these were not the ordinary Euclidean distances. The matrix created for this analysis had a 9.1% spatial connectivity and used the 8 nearest neighbors within the network for the GOG. The output of both approaches was interpreted based on the null hypothesis, whether the data were clustered, dispersed, or random.

## 4. Analysis and Results

### 4.1. Accident Severity, Contributory Causes, and Locations

The data obtained from the FRSC were analyzed to identify the fatal and injury-only accidents along the study highway. The results are shown in Table 1. As noted, 4656 accidents occurred within the study period (2013–2017). The majority of accidents occurred in 2013 when the number of accidents was 1285. This decreased to 861 accidents in 2014, 815 in 2015, 704 in 2016, and 991 in 2017. In 2013, the fatal accidents represented 40.5% of all fatal accidents recorded during the study period. However, 2015 and 2016 experienced a remarkable decrease in accident fatalities and injuries with 9.5 and 6.8% of fatal accidents, respectively. A sudden increase in the number of accidents/fatalities was experienced in 2017.

**Table 1.** Accident severity along Lokoja−Abuja-Kaduna highway.

| Year | No. of Accidents | Fatalities | | Injuries | |
|------|------------------|------------|---|----------|---|
| | | Frequency (F) | % | Frequency (F) | % |
| 2013 | 1285 | 1154 | 40.46 | 996 | 25.50 |
| 2014 | 861 | 818 | 28.68 | 767 | 19.64 |
| 2015 | 815 | 271 | 9.50 | 737 | 18.87 |
| 2016 | 704 | 195 | 6.84 | 621 | 15.90 |
| 2017 | 991 | 414 | 14.52 | 785 | 20.10 |
| Total | 4656 | 2852 | 100 | 3906 | 100 |

The obtained data were analyzed to identify the significant indicators for the specific safety problems at the locations. As noted in Table 2, speed violations (SPV) and loss of control (LOC) were the most common accident-contributing factors, comprising approximately 27% and 21% of all accidents, respectively. The two leading causes are interwoven, where drivers are liable to lose control of the steering at a high speed, resulting in an accident. The third leading cause of accidents is sign light violation (SLV), accounting for 16% of all accidents, followed by tyre bursts (TBT), which were approximately 11% of all accidents. Other factors that have not been reported account for less than 10% of the total accidents along the study route, which agreed with the annual report [48] and other published work [49]. This underreporting of accidents might be due to a lack of modern equipment for accident reports, poor reporting standards by officials, and a lack of adequate security. Moreso, accidents that happened late in the night (10:00–11:59 p.m.) or in the early hours (12:00–6:00 a.m.) of the day may not be reported as officials cannot be at all locations simultaneously.

**Table 2.** Contributory causes of road traffic accidents along the Lokoja−Abuja−Kaduna highway (2013–2017).

| | | 2013 | | 2014 | | 2015 | | 2016 | | 2017 | | Total | |
|---|---|---|---|---|---|---|---|---|---|---|---|---|---|
| No. | Contributory Cause | F | % | F | % | F | % | F | % | F | % | F | % |
| 1 | Speed Violation | 286 | 21.95 | 279 | 26.27 | 309 | 30.21 | 288 | 29.24 | 416 | 29.44 | 1578 | 27.27 |
| 2 | Loss of Control | 370 | 28.40 | 274 | 25.80 | 231 | 22.58 | 133 | 13.50 | 224 | 15.85 | 1232 | 21.29 |
| 3 | Sign Light Violation | 63 | 4.83 | 102 | 9.60 | 122 | 11.93 | 254 | 25.79 | 386 | 27.32 | 927 | 16.02 |
| 4 | Tyre Burst | 142 | 10.90 | 137 | 12.90 | 133 | 13.00 | 97 | 9.85 | 125 | 8.85 | 634 | 10.97 |
| 5 | Wrongful Overtaking | 184 | 14.12 | 62 | 5.84 | 44 | 4.30 | 25 | 2.54 | 45 | 3.18 | 360 | 6.22 |
| 6 | Dangerous Driving | 103 | 7.90 | 60 | 5.65 | 67 | 6.55 | 54 | 5.48 | 45 | 3.18 | 329 | 5.69 |
| 7 | Route Violation | 47 | 3.61 | 50 | 4.71 | 51 | 4.99 | 55 | 5.58 | 53 | 3.75 | 256 | 4.42 |
| 8 | Dangerous Overtaking | 27 | 2.07 | 19 | 1.79 | 08 | 0.78 | 07 | 0.71 | 23 | 1.63 | 84 | 1.45 |
| 9 | Mechanically Deficient Vehicle | 17 | 1.35 | 12 | 1.13 | 06 | 0.59 | 15 | 1.52 | 31 | 2.19 | 81 | 1.40 |
| 10 | Brake Failure | 08 | 0.61 | 22 | 2.07 | 16 | 1.56 | 10 | 1.02 | 15 | 1.06 | 71 | 1.23 |
| 11 | Others | 19 | 1.45 | 13 | 1.22 | 12 | 1.27 | 08 | 0.81 | 12 | 0.85 | 65 | 1.12 |
| 12 | Road Obstruction Violation | 10 | 0.76 | 14 | 1.32 | 12 | 1.17 | 13 | 1.32 | 13 | 0.92 | 62 | 1.07 |
| 13 | Fatigue | 09 | 0.69 | 04 | 0.38 | 03 | 0.29 | 18 | 1.83 | 16 | 1.13 | 50 | 0.86 |
| 14 | Driving under the Influence of Alcohol/Drugs | 09 | 0.69 | 05 | 0.47 | 05 | 0.49 | 03 | 0.30 | 03 | 0.21 | 25 | 0.43 |
| 15 | Overloading | 02 | 0.15 | 02 | 0.19 | 0 | 0 | 02 | 0.20 | 03 | 0.21 | 09 | 0.16 |
| 16 | Sleeping at the Wheel | 01 | 0.07 | 04 | 0.38 | 03 | 0.29 | 0 | 0 | 0 | 0 | 08 | 0.14 |
| 17 | Bad Road | 01 | 0.07 | 02 | 0.19 | 0 | 0 | 02 | 0.20 | 01 | 0.07 | 06 | 0.10 |
| 18 | Use of Phone While Driving | 02 | 0.15 | 01 | 0.09 | 0 | 0 | 01 | 0.10 | 01 | 0.07 | 05 | 0.09 |
| 19 | Poor Weather | 03 | 0.23 | 0 | 0 | 0 | 0 | 0 | 0 | 01 | 0.07 | 04 | 0.07 |
| | Total | 1303 | 100 | 1062 | 100 | 1022 | 100 | 985 | 100 | 1413 | 100 | 5786 | 100 |

Based on the accident locations, the study area was divided into four sections: Section I (Lokoja–Kotonkarifi), Section II (Kotonkarifi–Abaji), Section III (Abaji–Abuja), and Section IV (Abuja–Kaduna). Each section has routes and each route contains specific locations. In the northbound direction, 47 out of the 90 locations had accidents ten times or more, as shown in Table 3. The southbound direction consisted of 93 accident locations, of which 38 locations had accidents ten times or more, as shown in Table 4. These locations had more significant accidents between 2013 and 2017, emphasizing the need for further detailed analyses of these locations.

**Table 3.** Northbound locations with number of accidents of 10 or more (2013–2017).

| No. | Accident Location | Total Number of Accidents | No. | Accident Location | Total Number of Accidents |
|---|---|---|---|---|---|
| 1 | Gadabiyu town | 86 | 25 | Doka | 16 |
| 2 | Awawa | 53 | 26 | Idu Bridge | 15 |
| 3 | Manderegi | 52 | 27 | Giri Inter. | 15 |
| 4 | Banda | 48 | 28 | Rijana | 15 |
| 5 | Ahoko Village | 40 | 29 | Bako Village | 14 |
| 6 | Kara | 39 | 30 | FGC Kwali | 14 |
| 7 | Gwako Village | 33 | 31 | Anagada U-turn | 14 |
| 8 | General Hospital Inter. Kw | 28 | 32 | Azara Town | 14 |
| 9 | Okpaka | 26 | 33 | Kwaita | 13 |
| 10 | GSS Yangoji | 26 | 34 | Zuma Rock | 13 |
| 11 | NATACO Junct. | 24 | 35 | Gidan Busa | 13 |
| 12 | Small Sheda | 24 | 36 | Kwali Mrkt. U-turn | 12 |
| 13 | Gaba Hill | 22 | 37 | Opp. Coll. Of Edu. Zuba | 12 |
| 14 | SLAN F/ST | 21 | 38 | Madalla Inter. | 12 |
| 15 | OZI Village | 20 | 39 | KM14 DM Kurfi | 12 |
| 16 | KM85 Katari | 19 | 40 | Bishini Inter. | 12 |
| 17 | Ahoko bridge | 17 | 41 | Toll gate SBW | 11 |
| 18 | Aseni Village | 17 | 42 | Ohono | 10 |
| 19 | SDP Junct | 17 | 43 | Chikara Village | 10 |
| 20 | T/Maje U-turn | 17 | 44 | Fire Serv. Coll. Kwali | 10 |
| 21 | Akilibu | 17 | 45 | Zuba U-turn | 10 |
| 22 | Adabo Village | 16 | 46 | Polewire | 10 |
| 23 | Big Sheda U-turn | 16 | 47 | Maro | 10 |
| 24 | KM11 Murada | 16 | | | |

**Table 4.** Southbound locations with number of accidents of 10 or more (2013–2017).

| No. | Accident Location | Total Number of Accidents | No. | Accident Location | Total Number of Accidents |
|---|---|---|---|---|---|
| 1 | Chikara Village | 110 | 20 | Big Sheda U-turn | 17 |
| 2 | Kwaita | 108 | 21 | Giri Inter. | 17 |
| 3 | Piri | 94 | 22 | Rijana | 17 |
| 4 | Banda | 48 | 23 | Doka | 17 |
| 5 | T/Maje U-turn | 44 | 24 | M/M Bridge | 17 |
| 6 | Akilibu | 36 | 25 | GSS Yangoji | 16 |
| 7 | Omoko | 35 | 26 | Jamata Curve | 15 |
| 8 | Gadabiyu town | 34 | 27 | Awawa | 14 |
| 9 | Bako Village | 30 | 28 | Zuba U-turn | 14 |
| 10 | Anagada U-turn | 28 | 29 | Akpogu Village | 13 |
| 11 | Opp. Marist Coll. | 25 | 30 | Small Sheda by NASC | 13 |
| 12 | SLAN F/ST | 25 | 31 | Gwako Village | 13 |
| 13 | Aseni Village | 22 | 32 | Sabon Gari Gadabiyu | 12 |
| 14 | Gidan Busa | 22 | 33 | Okpaka | 12 |
| 15 | Bulletin | 21 | 34 | Zuba Inter. | 12 |
| 16 | Naharati | 21 | 35 | Dankogi | 11 |
| 17 | KM 85 Karari | 21 | 36 | Gen. Hospt. Inter. Kwali | 10 |
| 18 | Kotonkarifi | 18 | 37 | NNPC F/ST. | 10 |
| 19 | Opp. Coll. Of Edu. Zuba | 18 | 38 | KM 8 SBW | 10 |

Inter. = Intersection.

### 4.2. Spatial Distribution of Accidents

A field survey was conducted to identify the spatial distribution of accidents along the highway. The GPS coordinates of the affected locations were obtained. The GIS tools were used to show the accident locations on a digital map and analyze traffic accidents' hotspots. Weighted mean center, KDE, Moran's *I* Statistic, fishnet polygon, and network spatial weight matrix were used to show the spatial nature of the accident locations. All the accident data (2013–2017) were used for the mean center and density analysis. For the cluster analysis (Moran's *I* statistic), both the total and the yearly accident data were used. To allow for the comparison of results of the hotspot analysis, the whole accident data were used for the fishnet polygon analysis, while the spatial weight matrix used the whole and yearly accident data.

#### 4.2.1. Mean Center Analysis

Figure 2a illustrates the geographic mean center for the cumulative accident frequency (2013–2017) and the individual years. It can be observed that there was a shift in the concentration of road traffic accidents along the highway from Gadabiu town near the Dangara intersection in 2013 toward Yangoji town in 2014. A backward shift of the geographic mean center of accident frequency toward Fukafu town was observed for 2015. In 2016, the mean center shifted forward again. This time it was found in Sabon Gida town and 2017 recorded the maximum shift in the mean center of accident locations toward Rafin Pa near the airport intersection of the FCT road. The overall mean center is located somewhere midway between Sabon Gida and Yangoji on a highway curve. This suggests that highway intersections and curves somewhat influence accident occurrence.

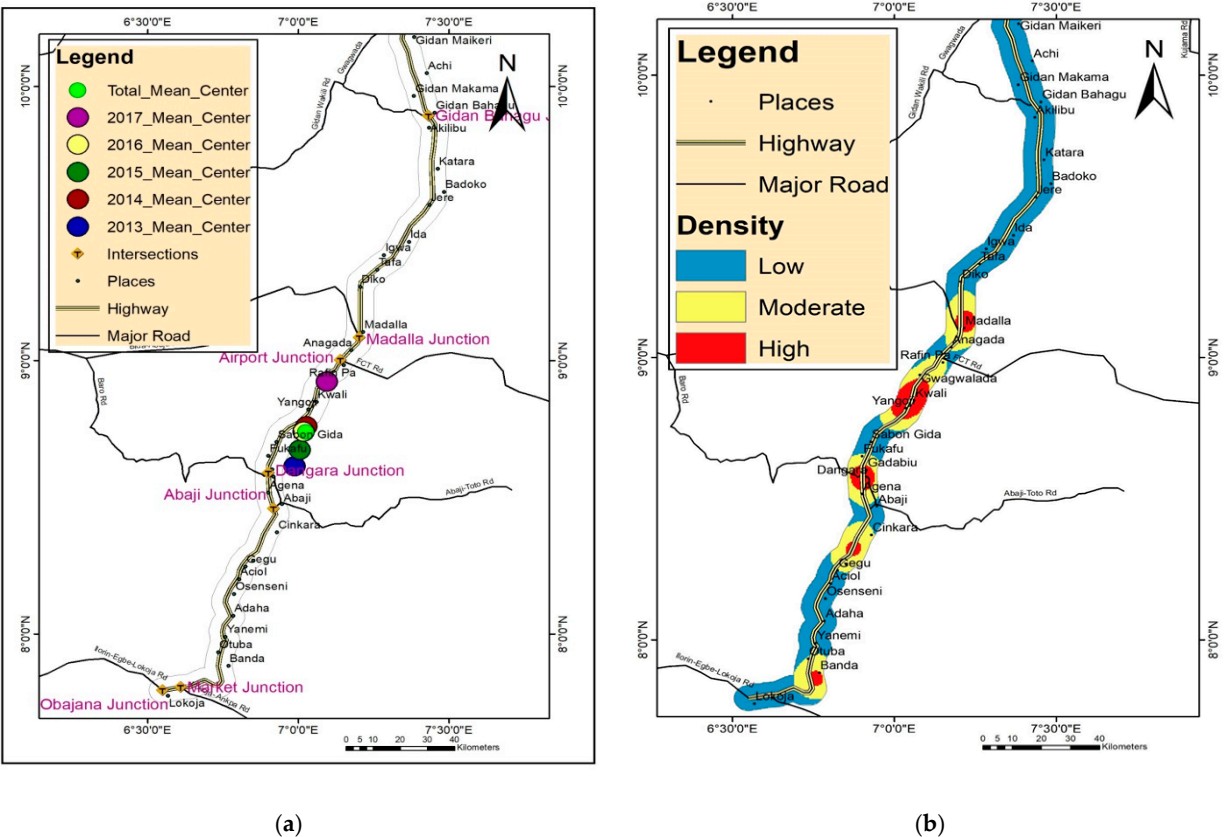

(**a**)                      (**b**)

**Figure 2.** Results of weighted mean and total density. (**a**) Shift in geographical mean of accident frequency; (**b**) Total hotspot map for 2013–2017.

### 4.2.2. Density Analysis

Figure 2b represents the KDE surface for the cumulative accident frequency across the study route. The density surface readily depicts the spatial variation in accident frequency from high to low. However, this is a subjective map, as it reveals nothing about the statistical significance of the high or low accident frequencies at locations across the study route. In other words, these variations could result from a random process and are not necessarily tied to a cause. Nevertheless, the density surface concerning the road network indicates that overall high frequency accident locations are associated with road intersections (e.g., Madalla and Dangara intersections) and the road curve near Banda town south of the study route.

### 4.2.3. Cluster Analysis

The results of the spatial autocorrelation (Moran's *I*) statistic of accident data for the five years are graphically illustrated in Figure 3a. The graph plots the Z-scores for each year against the other years. From the curve, it can be deduced that only in 2013, there was a significant cluster of high values of accident locations given by a ***Z***-score of 3.10538 with a less than 1% likelihood that the cluster could result from a random process; hence, the null hypothesis is rejected. The ***Z***-scores 1.7286 and 1.9496 for 2014 and 2017, respectively, indicate a lesser intensity of clustering with a 90% confidence interval and a 10% likelihood that the clusters could result from a random process. The ***Z***-scores for 2015 ($-0.459572$) and 2016 (0.182324) indicate that the pattern of accidents for the two years does not appear to be significantly different from random. However, Figure 3b, which represents the Moran's *I* result for the cumulative accidents, shows no overall accident clustering for the five years; the pattern is otherwise random.

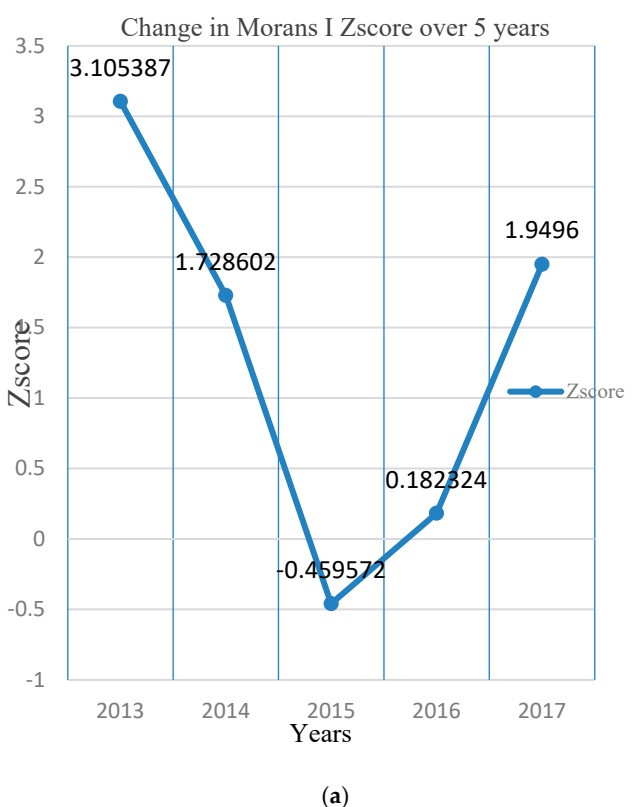

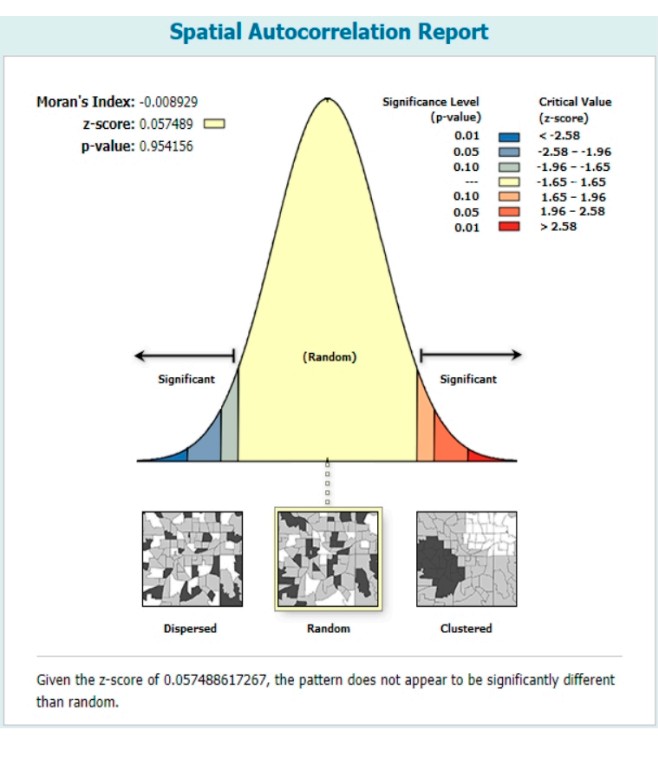

(**a**)

(**b**)

**Figure 3.** Line chart of *Z*-scores and the Moran's *I* total accidents report. (**a**) Change in Moran's *I* *Z*-score between 2013 and 2017; (**b**) Report of Moran's *I*: The total accidents for the 5 years.

4.2.4. Hotspot Analysis

Figure 4 represents the map output of the hotspot analysis using a fishnet polygon. The map demonstrates a statistically significant clustering of fishnet cells with a higher number of aggregated accident locations around the center of the study route closely associated with the Madalla and Airport intersections. It is noteworthy that the estimated hotspot is localized in a part of the study route where there is a relatively abrupt change in land elevation per unit distance on the highway. This may contribute to accident frequency in this hotspot-marked route.

The outputs of the hotspot analysis using a network spatial weight matrix for each year from 2013 to 2017 are represented in Figure 5A–E, respectively. There appears to be a shift in the hotspot location from the highway's south end in 2013 (at the road curve near the market intersection and Banda town) to the center of the study route in 2014 (near Dangara and Abaji intersections) and the north end of the highway in 2017 (near Gidan Bahagu intersection). There are, however, no hotspot locations for accidents recorded in 2015 and 2016. Also, the cumulative accident record for the five years expresses no hotspot locations, as shown in Figure 6. As shown in Figure 5A,B,E, the hotspot locations with high confidence levels are at points with geometric characteristics, such as intersections, curves, bridges, U-turns, interchanges, grades, hilly terrain, roadside obstacles, and median barriers.

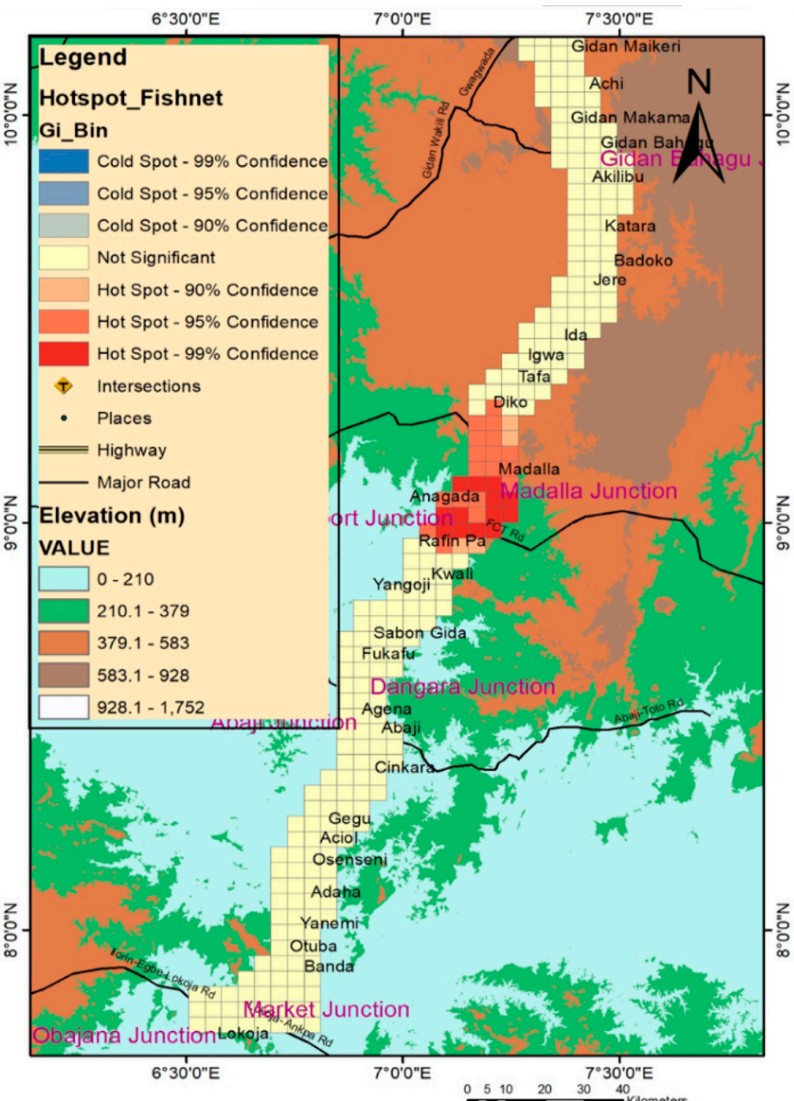

**Figure 4.** Accident hotspot map of the study route using a fishnet polygon bounded by the highway buffer polygon.

In 2013, based on the accident frequency at each of the 88 locations, the standard deviation measurements showed 17 locations with a **Z**-score above 2.0 standard deviations. Five of these 17 locations include the top 10 highest motor vehicle accident locations found to have a **Z**-score above 2.0 as determined by the hotspot analysis. In 2014, based on the accident frequency, the standard deviation measurements showed five locations with a Z-score above 2.0 standard deviations. Two of these five locations include values significantly higher than the other three within the neighborhood. Finally, in 2017, the standard deviation measurement found seven locations with a **Z**-score above 2.0 standard deviations. One of these seven locations stands out as a spot with less than a 1% likelihood that the clustering of road accidents results from a random process.

Therefore, if the decision were to be made based on these results, it would be better to look at the statistically significant clusters of high accident frequency for each year. Perhaps the shift in accident hotspots over the years is attributable to pavement failure (a common feature of Nigerian roads) and other factors, such as reckless driving, absence of or inadequate traffic signs, and vehicle worthiness.

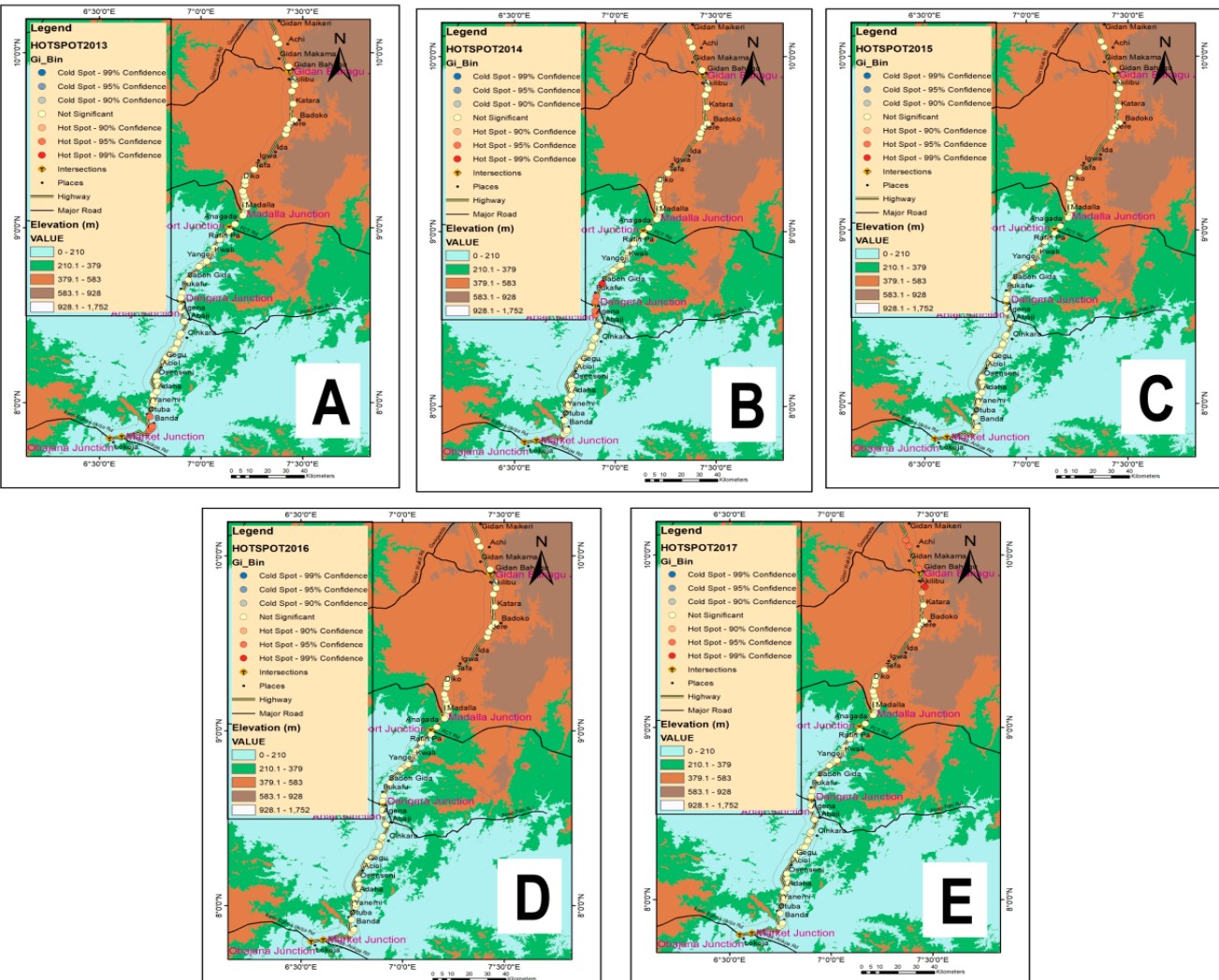

**Figure 5.** Hotspot analysis: (**A**) Year 2013: hotspot exits with a 95% significance, (**B**) Year 2014: hotspot exits with a significance between 95 and 99%, (**C**) Year 2015: no hotspot exits, the pattern is random, (**D**) Year 2016: no hotspot exits, the pattern is random, and (**E**) Year 2017: hotspot exits with a significance between 95 and 99%.

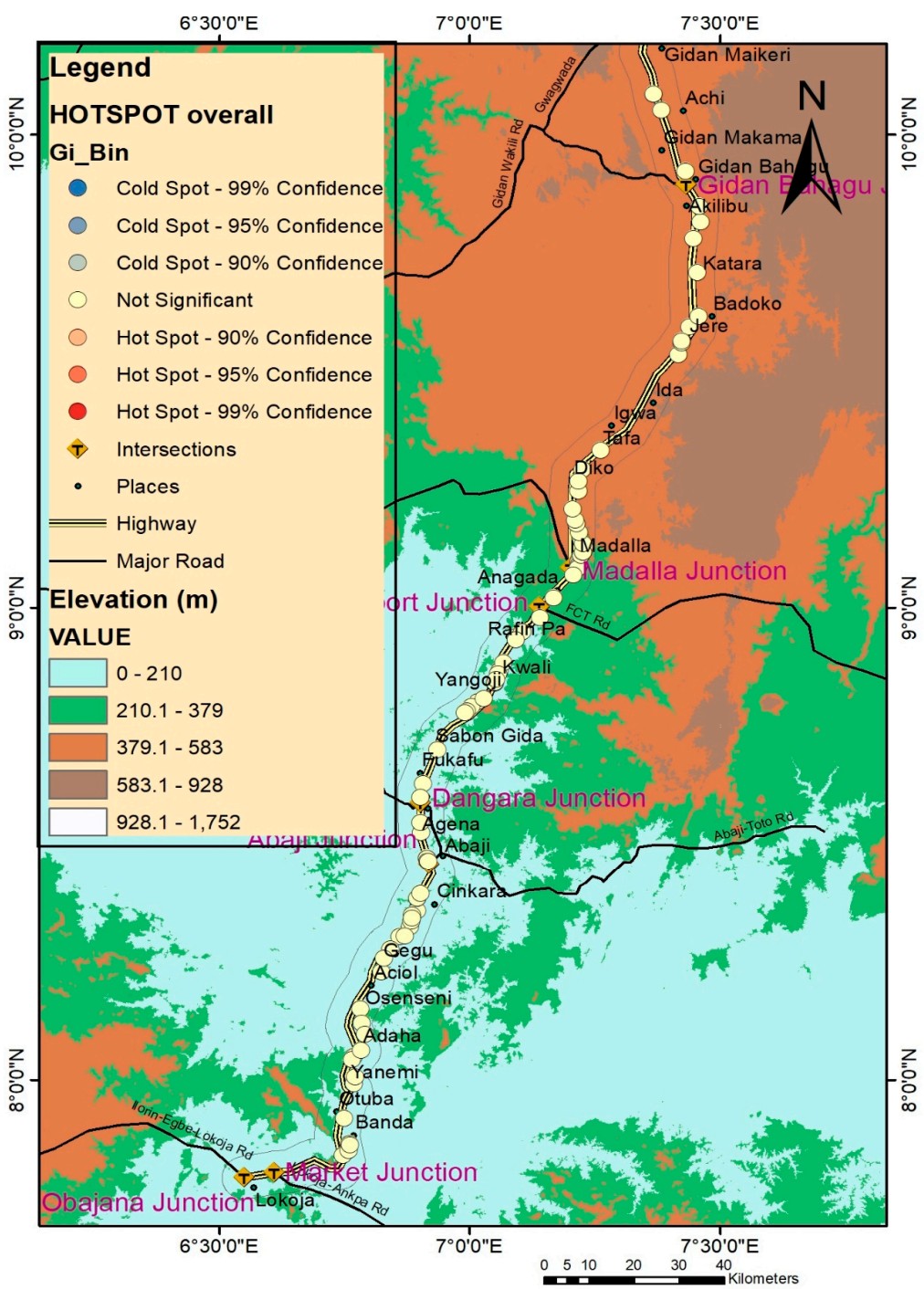

**Figure 6.** Overall hotspot analysis for the 5 years (2013 to 2017).

*4.3. Traffic Exposure*

Traffic counts were conducted to determine the effect of traffic volume at the hotspot locations on accident occurrence. Table 5 shows the traffic data for the sections where hotspots were identified. The locations within Abaji–Abuja have the highest ADT of 31,270 for the northbound direction and 16,303 for the southbound direction. Other sections have ADTS that are less than 10,000. As expected, the sections with larger traffic volumes have more accidents than those with lesser traffic volumes.

**Table 5.** Traffic data at hotspots.

| No. | Section | Direction [a] | Hotspot Locations | ADT |
|---|---|---|---|---|
| 1 | Lokoja–Kontokarifi | NB | Banda, Market Intersection, Karara | 4903 |
| | | SB | | 3836 |
| 2 | Kontokarifi–Abaji | NB | Sabon Gida, Agena, Pukafu, Dangara Intersection | 5600 |
| | | SB | | 4960 |
| 3 | Abaji–Abuja | NB | Abaji Bridge, Gen. Hospt. Intersection Abaji, Abaji U-turn | 31,270 |
| | | SB | | 16,303 |

[a] NB = Northbound direction, SB = Southbound zdirection.

### 4.4. Geometric Characteristics of Hotspots

Accident occurrences along the study route are not evenly distributed at the hotspot, as some occurred at locations with geometric features. From Table 6, most accidents occurred at horizontal curve locations, U-turns for villages and small cities, bridges, t-intersections, and roadside objects. Other accidents occurred at locations with a settlement, vertical curves, roadside parking, and eroded shoulders (Figure 7). These results agree with [50,51], who inferred that highway geometric features, roadside characteristics, and road design, among other factors, were the significant causes of road accidents in developed and developing countries. According to the FRSC report, high speeds at some hotspots are the primary cause of accidents at the locations. Driving at high speeds on sharp horizontal curves tends to result in accidents as the vehicle may swerve away from the road surface. Figure 8 shows the frequency of accidents at the hotspots identified for 2013, 2014, and 2017.

**Table 6.** Geometric characteristics of the hotspots identified for different years.

| Year [a] | Location | Geometric Characteristics [b] | Major Accident Causes [c] | C.L. (%) | Suggested Improvement |
|---|---|---|---|---|---|
| 2013 | Market Inter. | HC, Built-up area, eroded shoulder | High speed | 99 | Pedestrian bridge/parking lot |
| | Banda | HC, roadside obstacle (hill) | High speed | 99 | Speed limit |
| | Fukafu | HC, built-up area, | Sign violation | 99 | Proper signpost |
| | Dangara Inter. | Built-up area, U-turn, T-intersection | LOC | 95 | Proper signpost |
| 2014 | Agena | HC | High speed on sharp curve | 95 | Reconstruction |
| | Abaji Bridge | HC | Wrongful overtaking | 95 | Speed limit & signpost |
| | Gen. Hospt. Abaji | T-intersection, | High speed | 90 | Speed limit & signpost |
| | Abaji U-turn | U-turn | Fatigue | 99 | Reconstruction |
| | NAHARATI Abaji | U-turn, bridge, built-up area, vertical curve | LOC/pavement failure | 90 | Reconstruction |
| | Sabon Gida | HC, truck parking on shoulder & deceler. lane, U-turn | LOC | 99 | Proper road marking and signpost |
| 2017 | Achi | Vertical curve | LOC | 95 | Proper road marking and speed limit |
| | Gidan Bahagu | U-turn | Fatigue | 95 | Reconstruction |
| | Akilibu | Horizontal curve, T-intersection | Road obstruction | 99 | Intersection signalization |
| | Karara | Bridge, horizontal curve | High speed | 90 | Signpost required |

[a] No hotspots for 2015 and 2016, [b] HC = horizontal curve, [c] LOC = loss of control, C.L. = confidence level.

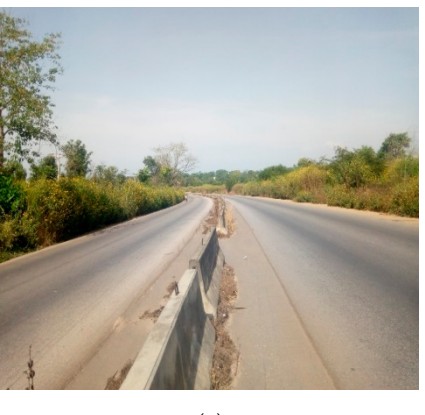

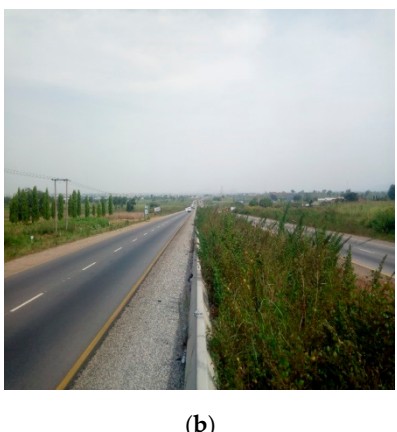

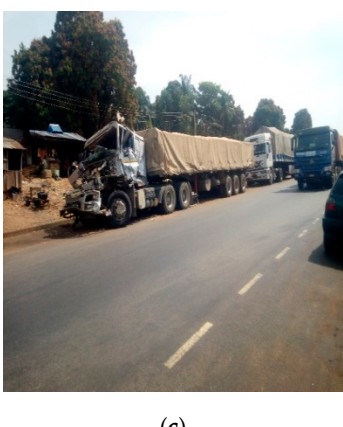

(**a**)                            (**b**)                            (**c**)

**Figure 7.** Hotspot locations along the study sections. (**a**) Sharp horizontal curve at Agena; (**b**) Vertical curve at Achi area; (**c**) Trucks parking on deceleration lane at Sabon Gida.

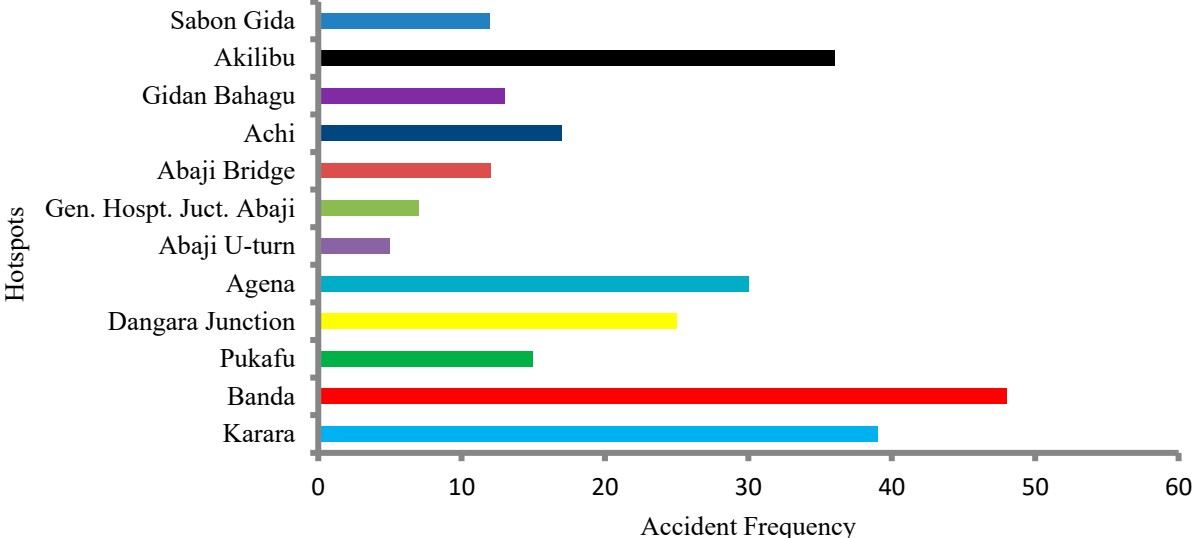

**Figure 8.** Accident frequency at the hotspots (2013, 2014, and 2017).

## 5. Discussion

The number of accidents per year at the network locations was identified based on subjective analysis of accident data. This method provides a primary indicator for the possible situation of particular safety problems at the locations. However, this unrealistic method can lead to false identification of hotspots and prioritization of the section's improvements. Moreover, this method is not in alignment with the criteria outlined by Overgaard [7] for hazardous road location identification.

The weighted mean center represented the concentration of road traffic accidents along the highway. This shows the shift in the geographical mean of accident frequency on the study route. The analysis indicated the overall mean center at the midway point between Sabon Gida and Yangoji on curves on the study route, as shown in Figure 2a. This justifies the findings by Paul [50], which inferred that geometric features, among other factors, are responsible for accident occurrences.

Figure 2b shows the KDE, which is highly responsible for visual detection. The KDE map shows the total frequency of accident locations related to road intersections such as Madalla and Dangara and road curves in Banda town. However, it only addresses the first-order properties in the hotspot spatial analysis of points without considering the spatial dependence and statistical significance of the interaction among the number of events in a given location [33].

The second-order effects of the spatial process were examined using GOG, which evaluates the extent to which a variable at a given location affects those of the neighboring locations [11]. As noted in Figure 3a, in 2013 only, there was a significant cluster of high accidents with a **Z**-score, *p*-value, and Moran's *I* index shown on Table 7. **Z**-scores, *p*-values, and Moran's *I* index for 2014 and 2017 indicate a lesser clustering intensity. Also, the values for 2015 and 2016, as indicated in Table 7, show that the accident pattern for the two years is random. The cumulative accidents with **Z**-score = 0.0575, *p*-value = 0.9542, and a Moran's *I* index of −0.0089 indicate that there is no overall clustering and that the accident occurrence for the five years does not appear to be significantly different from random.

**Table 7.** *Z*-scores for the years 2013–2017.

| Year | Z-Score | *p*-Value | Moran's *I* Index |
|------|---------|-----------|-------------------|
| 2013 | 3.1054 | 0.0019 | 0.1263 |
| 2014 | 1.7286 | 0.0839 | 0.0638 |
| 2015 | −0.4596 | 0.6458 | −0.0320 |
| 2016 | 0.1823 | 0.8553 | −0.0032 |
| 2017 | 1.9496 | 0.0512 | 0.0799 |

The hotspot analysis involved two approaches to obtain the GOG statistic: the fishnet polygon and network spatial weight matrix. The fishnet cells were statistically significant, with many accident locations around the Madalla and airport intersection in the study route. Land elevation per unit distance contributes to accident occurrences in the study route, as the estimated hotspot is localized in some areas. The network spatial weight matrix shown in Figure 5 indicates hotspot analyses for 2013 to 2017. There are shifts in the hotspot locations from one year to another. The hotspots exist for 2013, 2014, and 2017 with a 95–99% significance level. This occurs at locations with geometric features such as curves and intersections. The hotspots do not exist for 2015 and 2016 since the patterns are random. Also, the cumulative accident record for the five years shows no hotspots exist (Figure 6). Over the years, the shift in accident hotspots can be attributed to other causative factors (e.g., human, vehicle, and environmental factors).

The influence of traffic exposure on the hotspot locations is very significant in the Abaji–Abuja sections, which are comprised of the following hotspot locations: the Abaji bridge, the general hospital intersection Abaji, and the Abaji U-turn. An ADT of 31,270 was obtained for the NB direction and 16,303 for the SB direction. This high traffic exposure contributes to accidents at the hotspots as the section is a built-up area with commercial centers along the route. Other sections in the study route have a relatively low traffic volume, which is insignificant and does not influence the hotspots. In addition, the underreporting of accidents is envisaged to influence accident hotspot determination. The lack of adequate data capturing equipment, inexperience of officials, and unavailability of officials at all locations for 24 h a day and seven days a week might have resulted in some accidents not being captured.

## 6. Conclusions

This study has identified high-risk locations (hotspots), representing the first step in a safety improvement program. Accident locations from the primary and secondary data sources were mapped. The accident concentrations at the locations were determined using the weighted mean center and KDE methods. These locations were further verified using two different approaches to the GOG statistic (fishnet polygon and network spatial weight matrix). Based on this study, the following conclusions are made:

1. This study has contributed to the body of literature by showing the viability of the fishnet polygon and spatial weight matrix for the aggregation of accident locations and conceptualization of the spatial relationships among accident locations on a highway network. This is similar to the use of the SANET tool. The distance between features was measured within the network, rather than the ordinary Euclidean distances.

2. The concentration of road traffic accidents is midway between the Sabon-Gida and Yangoji curves, as indicated by the weighted mean center analysis. In addition, based on the visual detection conducted using KDE, the frequency of accident locations is associated with road intersections (such as the Madalla and Dangara intersections) and road curves in Banda town.

3. The hotspots exist with a significance level between 95–99% for 2013, 2014, and 2017. However, the cumulative hotspot map indicates that the pattern of hotspots for 2015 and 2016 is random. Thus, preventive measures for the hotspot locations should be based on a yearly hotspot analysis. Further, traffic exposure is significant at the accident hotspots of the Abaji Bridge, Gen. hospt. Abaji, and Abaji U-turn. Thus, precautionary measures should be put in place at these locations.

4. The spatial autocorrelation analysis of the overall accident locations with a $Z$-score = 0.0575, $p$-value = 0.9542, and Moran's $I$ statistic = −0.0089 showed that the distribution of accidents in the study route is random.

5. One limitation of the present study is that it did not include input variables such as pavement condition, grade, and sight distance in the analysis. Future research must examine such variables' influence in the analysis. In addition, future work is needed to check the consistency and reliability of highway geometric design features.

**Author Contributions:** Conceptualization, A.A.; Data collation, A.A., O.S.A., S.M.E., F.M.A. and O.F.; Formal analysis, A.A., O.S.A. and S.M.E.; Supervision, O.S.A., and S.M.E.; Writing—original draft, A.A., O.S.A., S.M.E., F.M.A. and O.F.; Interpretation and editing, A.A., O.S.A. and S.M.E. All authors have read and agreed to the published version of the manuscript.

**Funding:** This research did not receive any specific grant from funding agencies in the public, commercial, or not-for-profit sectors.

**Acknowledgments:** The authors are grateful to the Nigerian Federal Ministry of Works and Transport and the Federal Road Safety Commission for providing the data used in this study. The authors also thank Songnian Li for his valuable comments on an earlier version of this paper. Thanks also go to Lubi, Sylvester Powei for helping out with the GIS analysis. The authors declare that the content of this article has not been published previously. All the authors have contributed to the work described, read, and approved the contents for publication in this journal.

**Conflicts of Interest:** All the authors have no conflict of interest with the funding entity and any organization mentioned in this article in the past that may have influenced the conduct of this research and the findings. All the authors have been certified by their respective organizations for human subject research. The authors declare no conflict of interest.

## Abbreviations

The following abbreviations were used in this paper:

| | |
|---|---|
| ADT | Average daily traffic |
| DI | Dangerousness index |
| EB | Empirical Bayes |
| FMWT | Federal Ministry of Works and Transport |
| FRSC | Federal Road Safety Commission |
| GIS | Geographic information systems |
| GOG | Getis–Ord Gi* |
| GPS | Global positioning system |
| HC | Hierarchical clustering |
| KDE | Kernel density estimation |
| KDE+ | Extended KDE |
| NB | Northbound |
| SB | Southbound |
| STAA | Spatial traffic accident analysis |
| SANET | Spatial analysis along network |

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
