# Peer review of "GIS-Based Spatial Analysis of Accident Hotspots: A Nigerian Case Study"

_infrastructures, doi:10.3390/infrastructures7080103_

Round 1

Reviewer 1 Report

This paper is well written. However, several problems should be addressed:

(1) the data is old, can you add some data after 2017?

(2) Spatial analysis methods are not very complicated. Can you extract some new comprehensive solutions from existing approaches?

(3) Problems in the existing studies are not systematically explained in this paper. That is, the review of research status should be strengthened.

(4) The figures are not clear, In a large image, you need to delete some extra images to improve the clarity of the image.

(5) Your data is not real-time enough.  What would your study say if you included data from the last five years?  Or how much reference to the latest situation?

  •  

Reviewer 2 Report

This paper applied statistical methods to study high-risk locations for 15 the Lokoja-Abuja-Kaduna highway in Nigeria. The study is interesting and important as it serves as the first step to preventing accidents. My comments to further improve the paper are as follows:

The study concluded that for some years accidents are associated with road characteristics. However, the results also show no hotspots exist for other years. If the accidents are associated with road characteristics, then why have there been no hotspots for some years. Also why the hotspots are moving?

Reviewer 3 Report

The submitted manuscript describes the use of spatial analysis techniques applied to the evaluation of road safety on a freeway in Nigeria. It shows a large amount of work behind the text. Although the focus of the manuscript is country specific, it should provide a contribution to the general state of knowledge (i.e. road safety lessons to developing countries). An important drawback is the set of analysis techniques selected by the authors. A linear freeway does not benefit from 2D analysis as much as a road network.

Some feedback is provided below:

Concerning language, the manuscript would benefit from copyediting. I will enhance readability and interest to readers.

Abstract

Lines 23-24: The abstract should be kept succint. Instead of providing scores, you can just provide the bare interpretation.

Introduction

In the statement of the research problem, the authors should assert why they selected spatial analysis instead of usual generalized linear models. See comments below about additional variables that are not clear whether they were systematically considered.

Review of methods

Although the literature review comprises many documents, I missed the references below. The first one describes a method that would be very appropriate for the data of this study. The second and third ones contains the development of Moran’s Index and Getis Ord methods, which were used un the study.

Agüero-Valverde, J., and Jovanis, P. (2008) Analysis of Road Crash Frequency with Spatial Models. Transportation Research Record, 2061, 55–63

Moran, P. A. P. 1948. The interpretation of statistical maps. Journal of the Royal Statistical Society. Series B (Methodological) 10 (2): 243–251.

Getis, A. 2010. Spatial interaction and spatial autocorrelation: A cross-product approach. In Perspectives on Spatial Data Analysis. Berlin, Germany: Springer, 23–33.

Data collection and analysis:

Lines 296-299: Given the large variations of AADT showed in table 5, how did you determine that traffic exposure did not affect the number of accidents?

Line 308: The manuscript would benefit from adding theoretical background on mean center analysis, Kernel density and cluster analysis techniques. Actually, this content is better suited to section 2. To help enhance this part of the text, I refer the authors to the following book:

Loo, B.P.Y., and Anderson, T.K. (2016) Spatial analysis methods of road traffic collisions, CRC Press.

Line 321: Which features were considered besides location?

Line 338: why did the authors selected 7 km long segments? According to AASHTO highway safety manual segmentation must be done in a way that segments must cover lengths where the main features remain constant.

Line 366. Include authors’ name.

Results

Line 376. I guess the authors meant ‘preliminary analysis’

Line 384. Was this variable (pavement condition) somewhat incorporated into the analysis? If not, the conclusion might not be supported.

Line 401 - 404: Statement not clear. Why so many divisions?

Line 25: Were intersection locations incorporated in the analysis? If not, the conclusion might not be supported.

Line 46: In accident analysis, periods of 3 to 5 years are encouraged to be considered to avoid excessive randomness and better figure out accident patterns. The authors carried out the analysis with most techniques with one year period. This is probably the cause of some meagre conclusions.

Line 70: Was landform or other related variable such as highway grade or available sight distance somewhat incorporated into the analysis?

Discussion

Line 156: include authors’ name

Lines 172- 178. Include these data in a table.

Conclusion

Some conclusions, as drafted, seem contradictory among them because some spots are hazardous or not depending on the method used and, on the year analyzed.

What are the contributions of this study to knowledge.

Figures

The manuscript would benefit from figures (1 and 5) with higher resolution.

Thank you for giving me the opportunity to evaluate your paper.

Round 2

Reviewer 1 Report

This paper can be published as it is

Reviewer 3 Report

The authors have made a nice effort to improve the manuscript.

Despite the review, I would like to emphasize that the contribution of this study to the state of knowledge is still unclear.

Conclusions are not supported enough by the findings as the authors acknowledge that variables that would have supported them are no included in the analysis (pavement condition, grade, sight distance). At least the authors should acknowledge the limitations of the study.

As I mentioned in my previous review, periods of 3 to 5 years are encouraged to be considered in accident analysis to avoid excessive randomness and better identify accident patterns. Despite the authors incorporated five years data, these data were incorporated on a yearly basis in the analysis for most techniques.

Round 3

Reviewer 3 Report

The changes made are fair enough.